## Invited reply

computational biology/cognition/behaviour

**Authors for correspondence:**
T. Morita
e-mail: tmorita@alum.mit.edu
H. Koda
e-mail: koda.hiroki.7a@kyoto-u.ac.jp

# Difficulties in analysing animal song under formal language theory framework: comparison with metric-based model evaluation

## T. Morita and H. Koda

Primate Research Institute, Kyoto University, 41-2 Kanrin, Inuyama, Aichi 484-8506, Japan

ⓘ TM, 0000-0002-3900-5410; HK, 0000-0002-0927-3473

## 1. Introduction

This is a reply to a comment from De Santo & Rawski (DS&R) [1] regarding our recent investigation of the gibbon song syntax [2]. The major objectives are to clarify the (i) difference between the proposed analytical method and formal language theory (FLT) advocated by DS&R and (ii) difficulties in applying the existing FLT-based analysis to animal studies.

## 2. Difficulties in FLT analysis

Figure 1*a* depicts typical FLT-based human language analysis. Herein, the researchers prepare a mathematically defined *language*—a set of strings of symbols—termed $\hat{L}$ in the figure. It is essential to identify the strings that belong to $\hat{L}$ and those outside $\hat{L}$, *including those not observed*: for example, unbounded centre-embeddings—though never observed—are often assumed to exist [3].

Once $\hat{L}$ is well defined, the researchers search for the *smallest* class of languages containing $\hat{L}$. This search begins with smaller classes; the researchers check whether each class contains $\hat{L}$ using mathematical theorems [3,4].

The *idealization*, i.e. the inference of $\hat{L}$, based on available information, is the most challenging aspect of FLT-based analysis. In the absence of systematic procedures, idealization may result in loss of reproducibility and even fabrication of data. However, linguists idealize languages unsystematically—without a constructive algorithm or evaluation metric. Moreover, idealization is not induced solely from collections of uttered sentences (corpora). Instead, factors like interpretation and grammatical judgement of sentences—seldom available from

**Figure 1.** Schematic of research procedures in (*a*) FLT and (*b*) metric-based model comparison.

animal studies and often unreliable [5,6]—are used. Thus, without significant changes, idealization designed for human languages cannot be applied to animal data.

# 3. Non-FLT advantage of probabilistic context-free grammar

Based on difficulties due to idealization, it is more promising and practical to evaluate generative models of animal voice sequences from observed data, as depicted in figure 1*b*. This is noted to yield the best model among all hypothesized models; unlike FLT, it does not conclude that none of the models work. Thus, a broader range of models must be included for comparison, given an appropriate search algorithm. In [2], the hypothesis space for animal voice sequences was expanded to probabilistic context-free grammars (PCFGs) from previous regular domain. This difference between the FLT and proposed methods regarding search procedures has 'pushed the scientific community towards misguidance' with respect to idealization and *optimum-among-regulars* [7].

The *necessity* of models in a particular class is assessed using the proposed model comparison paradigm by defining the metric by its fit to data—*likelihood* under probabilistic settings. That is, a good model predicts the behaviour of real data with high probability and produces its generative simulation as realistically as possible. Natural language processing (NLP) adopts this metric; for example, the neural network parameters are optimized via likelihood maximization, which yields the current state-of-the-art language model [8–10]. Although the neural language modelling of animal voice sequences has not been studied extensively, it could become the empirical evidence for the necessity of superregular analysis if neural network models outperform the classical regular models [11–14]. The neural language model would also serve as the best animal voice sequence simulator currently available.

The likelihood metric is not suitable for non-neural, rule-based superregular models such as PCFG, as no remarkable advantages are exhibited over regular models for both human language [15] and animal voice sequences [2] (§4.2). Thus, NLP did not benefit much from superregularity prior to the deep learning era; the previous state-of-the-art architecture was smoothed $n$-gram models, which can only generate a subclass of regular languages (termed *strictly locally testable languages*) [16,17]. The results might appear to be counteractive, as the FLT proves that data with centre-embeddings can only be explained using superregular models. However, centre-embeddings are bounded and rare in real data [18]; therefore, they do not have a significant effect on statistical evaluation.

To study the advantages of superregular models, the *simplicity* of the models should be measured as well as the fit to data. The two submetrics can be balanced using the Bayesian posterior inference. A (non-regular) PCFG had greater posterior for human language data than regular grammars, which grow in size—decreasing the prior probability—to achieve the same degree of likelihood as the PCFG [15]. In [2], we showed that PCFG had the same advantage of compactness for analysis of gibbon song data (§4.3), to the extent that it outperformed regular grammars under the Bayesian metric (§4.1).

The compactness of PCFG probably arises from structural representation of frequent substrings. These statistical patterns are prominent even in child-directed speech [15] and animal voice sequences [19,20], where centre-embedding may not be observed. Improved versions of PCFG, such as *adaptor*

*grammars*, have been designed to better capture frequent substructures of sentences, rather than the centre-embeddings [21,22].

## 4. Execution costs

The processing of sentences using PCFG is generally based on a $O(n^3)$-time algorithm with respect to length of the sentence $n$ [23,24]; this method is polynomial but highly expensive for practical applications. The algorithm also requires a working memory of size $O(n^2)$, which eventually exceeds the capacity of human and animals. Contrarily, the finite-state automata run in linear time and the memory size remains constant. As noted by DS&R, differences in the costs of execution are not incorporated in the Bayesian analysis, which only focuses on Marr's computational level of inquiry and refrains from discussing the algorithmic or implementation levels [25].

It may be noted that biological organisms may not compute the exact probabilities as defined by PCFG. A reliable approximation with fading memory is acceptable in practice. For example, recurrent neural networks—including biologically plausible variants—act as universal approximators and run in real time [11,13]. Actual algorithms and implementation used by humans and animals are considered efficient but difficult to interpret, as in the case of neural networks. Hence, studies at the computational level help us understand human and animal cognitive systems upon investigation of their interpretable representations.

## 5. What is FLT expected to do for animal cognition studies?

Herein, the difficulties in applying FLT-based analysis to studies on animals have been identified. However, this does not mean that FLT is futile. For example, the discovery of more efficient algorithms is always valuable. Perhaps what is expected from FLT-oriented *linguists* is the proposal of a systematic idealization procedure that runs on real animal data. Various important achievements in FLT cannot be exploited unless this fundamental technology is developed.

Data accessibility. This article has no additional data.
Competing interests. We declare we have no competing interests.
Funding. This work was supported by the JSPS/MEXT KAKENHI (no. 4903(Evolinguistic), JP17H06380) and JST CREST no. 17941861 (no. JPMJCR17A4).

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
