## [Reviewer comments · Royal Society Open Science]

Review History

RSOS-192069.R0 (Original submission)

Review form: Reviewer 1

Is the manuscript scientifically sound in its present form?

Yes

Are the interpretations and conclusions justified by the results?

Yes

Is the language acceptable?

Yes

Do you have any ethical concerns with this paper?

No

Have you any concerns about statistical analyses in this paper?

No

Recommendation?

Accept with minor revision (please list in comments)

Comments to the Author(s)

With this reply to the commentary of De Santo & Rawski on their analysis of gibbon song syntax the authors clarify and defend their metric-based model evaluation. This clarification, making a comparison of the principles underlying FLT and their modelling approach, is certainly useful. The reply will not resolve the issue of the best or preferred way (and theoretical framework) for analyzing animal vocal sequences, but it will help to make readers aware of the differences in approaches and might stimulate further exploration of the topic. For this reason I am in favor of publication.

I realize that there is a constraint on the length of a reply. However, the explanation in the 2nd and 3rd paragraph of section 3 of the reply is quite dense and technical and hard to follow for someone not very familiar with the theoretical background of the various approaches. I would suggest that the authors might make this more accessible by expanding and if possible link it to a (hypothetical) case of animal vocal sequences. Doing so they may also make clear what type of empirical data are required to further explore and compare the value of the different approaches and the insights in the cognitive architecture that can be obtained by this.

Review form: Reviewer 2

Is the manuscript scientifically sound in its present form?

Yes

Are the interpretations and conclusions justified by the results?

Yes

Is the language acceptable?

Yes

Do you have any ethical concerns with this paper?

No

Have you any concerns about statistical analyses in this paper?

No

Recommendation?

Accept with minor revision (please list in comments)

Comments to the Author(s)

I find this short by reply level-headed and courteous, not looking for controversy but instead seeking to qualify the challenges in using FLT in realistic data, namely from non-human animals.

I don't have objections to their position that are worth mentioning in the context of this short reply.

I suggest citing and if found appropriate briefly discussing a recent paper by Linzen and Oseki, which discusses the reliability of acceptability judgments (<http://doi.org/10.5334/gjgl.528>). This could be done, for example, at the end of section 2. This would aid their position in that an important source of insight when idealizing a language - grammatical judgment - is not only "seldom available from animal studies", but also problematic in humans as well.

I caught a few minors errors:

Page 1, line 48: "In absence" should be "In the absence"

Page 2, line 41: "are assessed" should be "is assessed"

Page 3, line 23; "the Marr's" should be "Marr's"

Apart from what I've mentioned above, in my view this short piece should be published as is.

Decision letter (RSOS-192069.R0)

17-Dec-2019

Dear Dr Morita

On behalf of the Editors, I am pleased to inform you that your Manuscript RSOS-192069 entitled "Difficulties in Analyzing Animal Song under Formal Language Theory Framework: Comparison with Metric-Based Model Evaluation" has been accepted for publication in Royal Society Open Science subject to minor revision in accordance with the referee suggestions. Please find the referees' comments at the end of this email.

The reviewers and handling editors have recommended publication, but also suggest some minor revisions to your manuscript. Therefore, I invite you to respond to the comments and revise your manuscript.

- Ethics statement

- Data accessibility

If you wish to submit your supporting data or code to Dryad (<http://datadryad.org/>), or modify your current submission to dryad, please use the following link:
<http://datadryad.org/submit?journalID=RSOS&manu=RSOS-192069>

- Competing interests

- Authors' contributions

- Acknowledgements

- Funding statement

Because the schedule for publication is very tight, it is a condition of publication that you submit the revised version of your manuscript before 26-Dec-2019. Please note that the revision deadline will expire at 00.00am on this date. If you do not think you will be able to meet this date please let me know immediately.

- 1) A text file of the manuscript (tex, txt, rtf, docx or doc), references, tables (including captions) and figure captions. Do not upload a PDF as your "Main Document";
- 2) A separate electronic file of each figure (EPS or print-quality PDF preferred (either format should be produced directly from original creation package), or original software format);
- 3) Included a 100 word media summary of your paper when requested at submission. Please ensure you have entered correct contact details (email, institution and telephone) in your user account;

- 4) Included the raw data to support the claims made in your paper. You can either include your data as electronic supplementary material or upload to a repository and include the relevant doi within your manuscript. Make sure it is clear in your data accessibility statement how the data can be accessed;
- 5) All supplementary materials accompanying an accepted article will be treated as in their final form. Note that the Royal Society will neither edit nor typeset supplementary material and it will be hosted as provided. Please ensure that the supplementary material includes the paper details where possible (authors, article title, journal name).

If your manuscript is newly submitted and subsequently accepted for publication, you will be asked to pay the article processing charge, unless you request a waiver and this is approved by Royal Society Publishing. You can find out more about the charges at <https://royalsocietypublishing.org/rsos/charges>. Should you have any queries, please contact openscience@royalsociety.org.

on behalf of Dr Claudia Wascher (Associate Editor) and Kevin Padian (Subject Editor)
openscience@royalsociety.org

Associate Editor Comments to Author (Dr Claudia Wascher):

In their reply to a commentary of De Santo & Rawski (2019) the authors clarify and defend their approach of a metric-based model evaluation. Both reviewers find the reply valuable and recommend acceptance for publication after minor revisions.

Reviewer comments to Author:
Reviewer: 1

Comments to the Author(s)

With this reply to the commentary of De Santo & Rawski on their analysis of gibbon song syntax the authors clarify and defend their metric-based model evaluation. This clarification, making a comparison of the principles underlying FLT and their modelling approach, is certainly useful.

The reply will not resolve the issue of the best or preferred way (and theoretical framework) for analyzing animal vocal sequences, but it will help to make readers aware of the differences in approaches and might stimulate further exploration of the topic. For this reason I am in favor of publication.

I realize that there is a constraint on the length of a reply. However, the explanation in the 2nd and 3rd paragraph of section 3 of the reply is quite dense and technical and hard to follow for someone not very familiar with the theoretical background of the various approaches. I would suggest that the authors might make this more accessible by expanding and if possible link it to a (hypothetical) case of animal vocal sequences. Doing so they may also make clear what type of empirical data are required to further explore and compare the value of the different approaches and the insights in the cognitive architecture that can be obtained by this.

Reviewer: 2

Comments to the Author(s)

I find this short by reply level-headed and courteous, not looking for controversy but instead seeking to qualify the challenges in using FLT in realistic data, namely from non-human animals.

I don't have objections to their position that are worth mentioning in the context of this short reply.

I suggest citing and if found appropriate briefly discussing a recent paper by Linzen and Oseki, which discusses the reliability of acceptability judgments (<http://doi.org/10.5334/gjgl.528>). This could be done, for example, at the end of section 2. This would aid their position in that an important source of insight when idealizing a language - grammatical judgment - is not only "seldom available from animal studies", but also problematic in humans as well.

I caught a few minors errors:

Page 1, line 48: "In absence" should be "In the absence"

Page 2, line 41: "are assessed" should be "is assessed"

Page 3, line 23; "the Marr's" should be "Marr's"

Apart from what I've mentioned above, in my view this short piece should be published as is.

Author's Response to Decision Letter for (RSOS-192069.R0)

See Appendix A.

Decision letter (RSOS-192069.R1)

09-Jan-2020

Dear Dr Morita,

It is a pleasure to accept your manuscript entitled "Difficulties in Analyzing Animal Song under Formal Language Theory Framework: Comparison with Metric-Based Model Evaluation" in its current form for publication in Royal Society Open Science.

You can expect to receive a proof of your article in the near future. Please contact the editorial

office (openscience_proofs@royalsociety.org) and the production office (openscience@royalsociety.org) to let us know if you are likely to be away from e-mail contact -- if you are going to be away, please nominate a co-author (if available) to manage the proofing process, and ensure they are copied into your email to the journal.

Kind regards,
Lianne Parkhouse
Royal Society Open Science
openscience@royalsociety.org

on behalf of Dr Claudia Wascher (Associate Editor) and Kevin Padian (Subject Editor)
openscience@royalsociety.org

Appendix A

Responses to the Comments from the Reviewers on ID RSOS-192069, "Difficulties in Analyzing Animal Song under Formal Language Theory Framework: Comparison with Metric-Based Model Evaluation"

Takashi Morita

Hiroki Koda

Dear Dr. Claudia Wascher, our handling editor of Royal Society Open Science,

We would like to thank you for accepting our manuscript for publication in Royal Society Open Science.

We made minor revisions (red-colored) to the manuscript in accordance with the suggestions raised by the reviewers.

Best regards,

Takashi Morita, on behalf of the coauthor

Responses to the Reviewers' comments:

Reviewer: 1

With this reply to the commentary of De Santo & Rawski on their analysis of gibbon song syntax the authors clarify and defend their metric-based model evaluation. This clarification, making a comparison of the principles underlying FLT and their modelling approach, is certainly useful. The reply will not resolve the issue of the best or preferred way (and theoretical framework) for analyzing animal vocal sequences, but it will help to make readers aware of the differences in approaches and might stimulate further exploration of the topic. For this reason I am in favor of publication.

Thank you so much for your appreciation of our work.

I realize that there is a constraint on the length of a reply. However, the explanation in the 2nd and 3rd paragraph of section 3 of the reply is quite dense and technical and hard to follow for someone not very familiar with the theoretical background of the various approaches. I would suggest that the authors might make this more accessible by expanding and if possible link it to a (hypothetical) case of animal vocal sequences. Doing so they may also make clear what type of empirical data are required to further explore and compare the value of the different approaches and the insights in the cognitive architecture that can be obtained by this.

We were allowed to extend the manuscript a bit and clarified the two paragraphs at issue. In the 2nd paragraph of Section 3, we explained the likelihood/fit-to-data metric in prose in the second sentence. We also clarified how neural networks could be used for the animal studies.

In the 3rd paragraph, we clarified what the "n-gram models" (= the previous state-of-the-art models) are, by pointing out that they are a part of the regular models (and thus cannot generate/predict the unbounded center-embeddings, which FLT considers as the important aspect of the idealized human language). We think this helps readers follow the discussion in the paragraph.

Reviewer: 2

I find this short by reply level-headed and courteous, not looking for controversy but instead seeking to qualify the challenges in using FLT in realistic data, namely from non-human animals. I don't have objections to their position that are worth mentioning in the context of this short reply.

Thank you for your appreciation of our work.

I suggest citing and if found appropriate briefly discussing a recent paper by Linzen and Oseki, which discusses the reliability of acceptability judgments (<http://doi.org/10.5334/gjgl.528>). This could be done, for example, at the end of section 2. This would aid their position in that an important source of insight when idealizing a language - grammatical judgment - is not only "seldom available from animal studies", but also problematic in humans as well.

Thank you for letting us know this work. We included it in our references together with Gibson & Fedorenko (2010). The word limit is quite tight, so we are only able to refer briefly in the main text to the controversy in reliability of the grammatical judgement (in the second last sentence of Section 2).

I caught a few minors errors: Page 1, line 48: "In absence" should be "In the absence" Page 2, line 41: "are assessed" should be "is assessed" Page 3, line 23; "the Marr's" should be "Marr's"

Thank you for finding those typos. We corrected all of them.